# Social contact as a strategy to reduce stigma in low- and middle-income countries: A systematic review and expert perspectives

Carlijn Damsté[1☯], Petra C. Gronholm[2‡], Tjitske de Groot[3‡], Dristy Gurung[2,4‡], Akerke Makhmud[5,6‡], Ruth M. H. Peters[7‡], Kim Hartog[8,9☯]*

1 IQ Health, Radboud University Medical Center, Nijmegen, The Netherlands, 2 Health Service and Population Research Department, Institute of Psychiatry, Psychology and Neuroscience, Centre for Global Mental Health and Centre for Implementation Science, King's College London, United Kingdom, 3 Department of Development & Education of Youth in Diverse Societies, Utrecht University, Utrecht, The Netherlands, 4 Transcultural Psychosocial Organization (TPO) Nepal, Kathmandu, Nepal, 5 Centre for Global Mental Health, Institute of Psychiatry, Psychology and Neuroscience, King's College London, London, United Kingdom, 6 Rapid Research Evaluation and Appraisal Lab (RREAL), University College London, London, United Kingdom, 7 Faculty of Science, Athena Institute, Vrije Universiteit Amsterdam, Amsterdam, The Netherlands, 8 Department of Research and Development, War Child, Amsterdam, The Netherlands, 9 Amsterdam Institute for Social Science Research, University of Amsterdam, Amsterdam, The Netherlands

☯ These authors contributed equally to this work.
‡ These authors also contributed equally to this work.
* kim.hartog@warchild.nl

**Data Availability Statement:** The data underlying this study are available on Figshare repository

## Abstract

Social contact (SC) has been identified as a promising strategy for stigma reduction. Different types of SC exist. Various scholars defined positive factors to strengthen SC. This study aims to investigate the application and effectiveness of SC as a strategy to reduce stigmatisation across stigmas, settings and populations in low- and middle-income countries (LMICs). We specifically examine the use of positive factors. A systematic review was conducted in twelve electronic databases using key terms related to stigma AND social contact AND intervention AND LMICs. Data were synthesised narratively. Study quality was assessed with the Joanna Briggs Institute critical appraisal checklists. Additionally, semi-structured interviews were used with first/corresponding authors of included publications to investigate their practical experiences with SC. Forty-four studies (55 publications) were identified. Various stigmas (n = 16) were targeted, including mental health (43%). Indirect (n = 18) and direct contact (n = 16) were used most frequently, followed by collaboration, imagined and vicarious contact, or a combination. The most applied additional strategy was education. Almost half of the studies, explicitly or implicitly, described positive factors for SC, such as PWLE training or disconfirming stereotypes. The majority suggested that SC is effective in reducing stigma, although inconsistent reporting overshadows conclusions. Perspectives of people with lived experience (PWLE) were infrequently included. Expert perspectives stressed the importance of contextualisation, PWLE participation, and evaluation of SC. This study provides an overview of SC as a stigma reduction strategy within LMICs. Conclusions about which type of SC is more effective or whether SC is more effective for a specific stigma category cannot be drawn. We recommend future research to strengthen

(Link: https://figshare.com/s/
5bd0d2090406c501baba).

**Funding:** PCG is supported by the UK Medical
Research Council (UKRI) for the Indigo Partnership
(MR/R023697/1) award. CD has conducted this
work as a student and thereafter voluntary next to
an unrelated job. KH is funded by War Child
Holland's unearmarked funds. The funders had no
role in study design, data collection and analysis,
decision to publish, or preparation of the
manuscript.

**Competing interests:** The authors have declared
that no competing interests exist.

reporting on effectiveness as well as PWLE perspective and SC processes, and to further
critically examine the potential of SC. An overview of positive factors applied to strengthen
SC is provided, which can stimulate reflection and guide future SC.

## Introduction

Stigma is well-known to have a profound negative impact on health and quality of life, and the
construct has been of interest to various scholars since Goffman's seminal work [1]. Stigma
was further conceptualised as a phenomenon rooted in social interaction, defined as "the co-
occurrence of (. . .) labelling, stereotyping, separation, status loss, and discrimination" and
specified that "for stigmatization to occur, power must be exercised" [2].

Stigmatisation limits access to services (including, but not limited to, health) and engage-
ment in care [3], poses a barrier to help-seeking behaviours [4], negatively influences social
relationships and participation [5], reduces the opportunities of individuals [1] including
access to resources [5], and in general contributes to health inequity and social inequalities
[5,6]. Stigma could cause more harm than the burden of the condition itself [7–10]. At the
intersection of multiple stigmas, the (health) impact could be compounded [11,12].

The detrimental burden of stigmatisation on population health demands action. Recent
reviews investigated the state of the art of stigma reduction interventions [7,9,13–16]. Com-
pared to high-income countries (HIC), development and evaluations of anti-stigma programs
is limited in low- and middle-income countries (LMICs) [17,18]. As interventions are context-
dependent, those originating from HIC cannot be automatically transferred to LMICs [19].

Social contact (SC) has been identified as a promising strategy for stigma reduction
[7,18,20] which we operationalise as intentional interaction between people with lived experi-
ence of a certain (stigmatised) condition (PWLE) and people without that specific condition
[21–24]. Different SC types exist, such as direct, indirect and imagined contact. The rationale
of direct face-to-face SC originated from the perspective that contact between majority and
minority groups could reduce prejudice [25,26]. In situations where direct contact is less appli-
cable due to e.g. presence of high prejudice [27] or access restrictions [28], SC types such as
indirect (i.e. non-face-to-face contact such as video testimonials or radio diaries [27]), imag-
ined (i.e. imagining positive interaction [29]), vicarious (i.e. observing in-group members hav-
ing successful cross-group contact [30]), and extended (i.e. knowing that in-group members
have cross-group friends [31]) SC approaches have also been increasingly and successfully
applied [27,28]. Moreover, these SC types are often used to reach out to large audiences as they
are easy to spread and easy to scale up, such as in (large) campaigns [32,33].

The application of SC as a stigma reduction strategy comes with a few knowledge gaps.
First, recent systematic reviews or frameworks on SC as a stigma reduction strategy focused on
mental health stigma and indirect SC only [28,34,35], although SC has been employed to
reduce physical health stigma e.g. HIV/AIDS [32,36] and not health-related stigma concerning
age [37] or the experience of sexual violence [38]. Recent research advocates to learn about
stigma (reduction) across stigmas [12,39]. Second, although several scholars have investigated
which (combination of) positive factors–referred to as "optimal conditions" [21,25,26] or key
ingredients [40]–are required for SC to be more effective and least harmful to reduce prejudice
and stigmatisation, researchers have indicated more knowledge is required to improve SC in
practice [20,36,41]. Third, recent research highlighted that the evidence-base of SC is con-
tested, for example through biased reporting and lacking methodological rigor [23]. Addition-
ally, there are several criticisms, such as that SC may enhance rather than reduce

stigmatisation [24,36,42], or that positive testimonies of PWLE might not be believed and therefore increase stereotypes [43]. The above gaps trigger a more thorough look into the application and effectiveness of SC.

Against this background, to contribute to the knowledgebase, this study investigates SC as a strategy to reduce stigmatisation across health-related and not health-related stigmas, populations and settings in LMICs. The main aim of the systematic review was to identify contact-based stigma reduction interventions used in LMICs, across stigmas and populations, and assess their content and effectiveness.

To support the assessment in content and effectiveness, we additionally aimed to:

a. Examine whether, and if so which, known or new factors to strengthen SC have been applied; and

b. Explore which lessons were drawn to improve SC

These questions were answered by the review and complemented with expert perspectives.

To support the use of this review and stimulate reflection on and guide future implementation of SC, we have summarised these findings and recommended future research directions to improve SC.

## Materials and methods

A systematic review (part 1) and an additional exploration of expert perspectives (part 2) were conducted.

### Part 1: Systematic review

This review followed the Preferred Reporting Items for Systematic Reviews and Meta-Analyses (PRISMA) statement [44]. A protocol was developed a priori and registered on PROSPERO (ID: CRD42022311676). The PRISMA checklist can be found in **S2 Checklist**.

**Search strategy and study selection.**    Twelve electronic databases (Academic Search Premier, Anthropology Plus, CINAHL, Cochrane Library, Embase, ERIC, PsycINFO, PubMed, Scopus, SocINDEX, Sociological Abstracts, and Web of Science) were searched in February 2022 and repeated in February 2023. The search strategy included key terms related to stigma AND social contact AND intervention AND LMICs as defined by the World Bank classification list [45]. In the medical databases, the strategy was supplemented with medical subject headings (MeSH) [46]. The complete search strategy is provided in **S1 Text**. Additional search strategies were performed: 1) identified reviews and included studies were cross-referenced, and 2) first/corresponding authors of included studies were contacted for additional relevant studies.

Initial screening was conducted based on title and abstract. For studies considered relevant, full texts were assessed and screened against the eligibility criteria. "Covidence–Better systematic review management" software was used. Both during the initial and full text screening phase, 20% of the records were independently reviewed by two researchers (CD, KH). Discrepancies were discussed until consensus was reached. In case of ≥5% disagreement about inclusion, this process was repeated. The inter-rater reliability scores of the title/abstract and full text screening were 90% and 92% respectively.

**Eligibility criteria.**    Studies were included in this review if they 1) were a peer-reviewed article reporting primary research, 2) were situated in a LMIC according to the World Bank classification (2020), 3) described an intervention in which SC is used as a *strategy for stigma reduction among the population without the stigma to address*, 4) assessed stigma reduction quantitatively and/or qualitatively, and 5) were written in English, Dutch, French, Spanish, or German. There were no restrictions on stigmatised characteristics nor publication date. An

intervention in which SC is used as a strategy was understood as any form of created contact where PWLE and people without that stigma experience interacted together, through any form of SC. Stigma reduction was understood as a change in stigmatising practices or experiences, which might be reported in different ways such as increased warmth/empathy or reduced social distance. Studies were excluded when SC was not explicitly initiated as part of a stigma reduction intervention, such as general social media exposure or existing interactions. Studies were also excluded in case of two-way prejudice, implying that there was no strict power imbalance and thereby did not meet the stigma definition used [2]. A list of all inclusion/exclusion criteria is provided in S1 Text.

**Data extraction and quality assessment.** Data from the included studies were extracted in Excel and included general information about the study (e.g. publication year, author name, country), study methods, participant characteristics, type of intervention (including SC) and its content, information regarding positive factors, and effectiveness. The development of the extraction sheet was informed by previous work [14], after which it was pilot-tested and adjusted where necessary. Data extraction was independently conducted by two researchers (first and last author). Inter-rater reliability scores were high (85%, 90%, 96%; for 3 studies), thus the remaining studies were divided between the two researchers (first and last author). All studies were cross-checked by the first of last author for accuracy to increase internal validity. Discrepancies were resolved through discussion.

The quality of included studies was assessed by two researchers (CD, KH): both checked 15% independently. As discrepancies were minimal, the remaining studies were divided among both researchers. Arising questions were discussed and resolved. Joanna Briggs Institute (JBI) critical appraisal checklists were used that were specific to the research methodology [47]. In case a study used mixed methods related to stigma outcomes, both a quantitative and qualitative checklist was used. *High* quality was defined when ≥85% of relevant questions of the JBI checklist could be answered with a "yes", *moderate* quality when this was the case for 40%-<85% of relevant questions, and *low* quality when this was the case for <40%.

**Data synthesis.** As both quantitative and qualitative studies were included, JBI's convergent integrated approach for mixed methods reviews was used [48]. Narrative synthesis was conducted. Subsequent inductive qualitative analysis of extracted data on positive factors and lessons learnt was conducted, and themes and categories were refined through discussion. Specific attention was given to stigma; age cohort; location; and effectiveness per social contact type, supported by visualisations.

## Part 2: Expert perspectives

To provide additional insight into the application of SC as a stigma reduction strategy, expert perspectives were explored through interviews. The aim was to enrich the systematic review with additional insights into the application of SC as a stigma reduction strategy, such as choices around the type of SC and positive factors for SC. Reporting of this qualitative research followed Consolidated Criteria for Reporting Qualitative Research (COREQ) guidelines [49].

**Ethics.** A protocol was developed a priori. Ethical approval was requested from the Research Ethics Committee Arnhem-Nijmegen (registration number: 2022–13787); the committee judged that ethical approval was not required under Dutch National Law. Complete voluntary participation was stressed in the written information and repeated at the beginning of each interview. Each respondent provided informed consent.

**Study design.** Semi-structured individual interviews were conducted with first/corresponding authors of studies included in the review. All interviews took place on-line in May and June 2022.

**Recruitment strategy and respondents.**   The qualitative study was announced in the same e-mail in which first/corresponding authors of included studies were asked for additional relevant studies. This announcement was followed by an official invitation to participate. All respondents received written information about the project. Purposive sampling was used to ensure diversity, with several considerations guiding the recruitment: 1) approach the authors of most recent publications first, 2) variety of SC types, 3) variety of stigmas and contexts, 4) studies provided a rationale to apply SC, and 5) SC was the main stigma reduction strategy. Respondents needed to speak English or Dutch. In case of no response after contacting twice, first/corresponding authors, who had not been approached as they did not meet all abovementioned considerations, were contacted. We aimed to interview at least 10% of first/corresponding authors of included studies in the review. This 10% was based on a practical reason of available time; it was expected not to counteract the explorative purpose of the interviews.

**Data collection.**   A semi-structured topic guide was created and pilot-tested (see **S2 Text**). During the interview, experiences of stigma reduction researchers/practitioners concerning use of SC were collected. Each interview started with introducing the researcher, explaining the goal of the interview, and re-confirming informed consent. Interviews were held by a trained interviewer (first author) through a video call with Microsoft Teams. The interviewer had no (work) relation with any of the participants. All interviews were recorded and transcribed a verbatim in the language spoken during the interview. Anonymity of the participants was maintained during analysis. Data were stored in a password-protected secure database.

**Data analysis.**   Transcripts were analysed using thematic analysis, a qualitative research method "for identifying, analysing and reporting patterns within data" [50]. Qualitative software programme NVivo 12 software was used. The first transcript was independently coded by two researchers (first and last author) to minimise subjectivity. The findings were discussed, and discrepancies debated until consensus was reached. As discrepancies were minimal, the remaining transcripts were coded by the first author and checked by the last author.

## Results

### Systematic review

**Study selection.**   The first search identified 2686 records, of which 889 were duplicates. The second search identified 276 records with 60 duplicates. Through title/abstract screening of the 2013 remaining records, 1739 were considered irrelevant. After full-text screening, 244 records were excluded. Additional search strategies identified 25 eligible records. Eight unique studies had two or more publications. This resulted in a total of 44 main studies with 55 underlying publications included in this review. We described the results based on the 44 main studies and, in case of multiple publications, supplemented information, when necessary, from the corresponding publications. **Fig 1** presents a PRISMA flow diagram of the screening and selection process.

**General study characteristics.**   Key characteristics for each study are provided in **Table 1**. Publications run from 2003 to 2022. Studies took place in 19 countries covering all WHO regions. Most studies were conducted in the South-East Asian region (n = 14, 32%), the European (n = 10, 23%), African (n = 9, 23%) and Western Pacific (n = 7, 16%) regions. The Eastern Mediterranean and Americas regions were underrepresented with two studies (5%) each. Studies were randomised controlled trials (RCTs) (n = 16, 36%), quasi-experimental (n = 15, 34%) or non-comparison studies (n = 13, 31%).

**General intervention characteristics.**   The 44 included studies had together a total of 84 study arms. In total, 53 study arms included SC, as nine control arms included a comparison stigma reduction intervention with SC. The other arms consisted of 15 control arms without

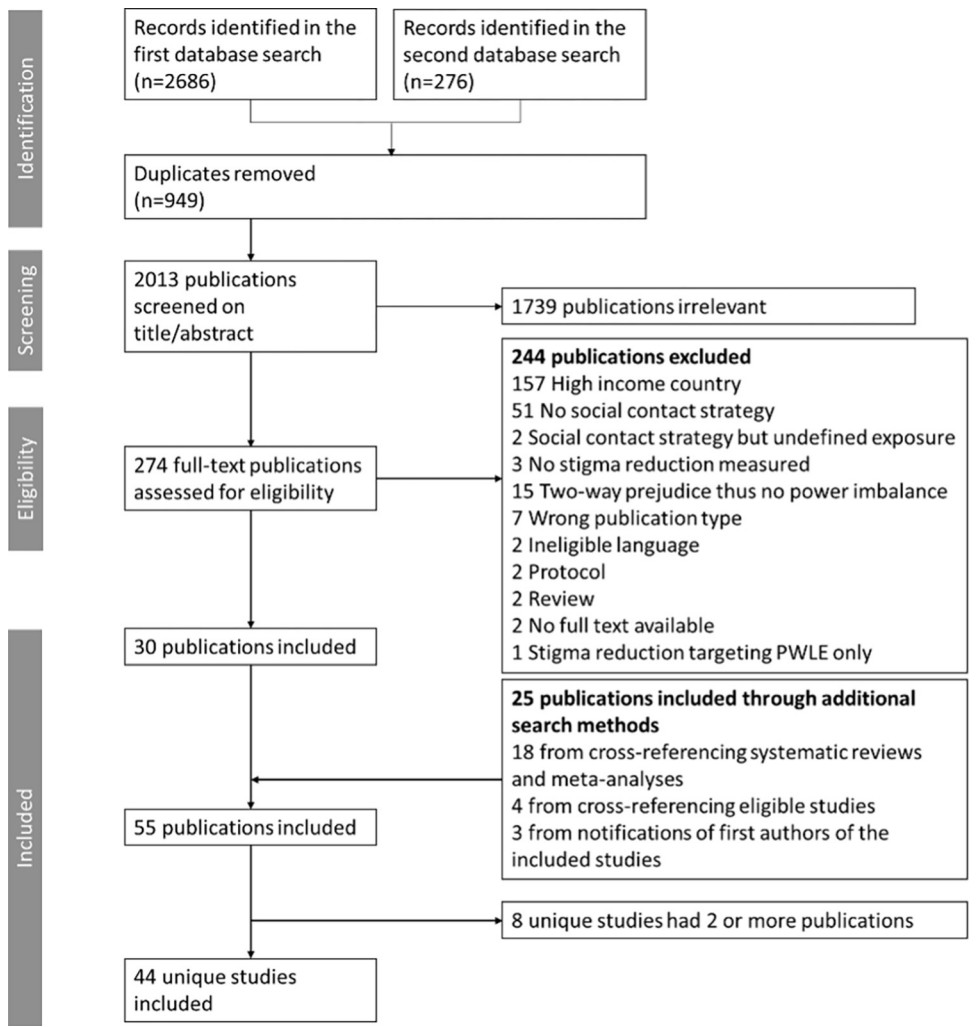

**Fig 1. PRISMA flowchart.**

an intervention, 11 with a comparison stigma reduction intervention without SC, and 5 with an intervention irrelevant to stigma reduction. Of the fifty-three SC interventions, 18 (34%) employed indirect and 16 (30%) direct SC, 8 (15%) used a combination of SC approaches, 7 (13%) employed SC through collaborative activities, and 4 (8%) employed imagined or vicarious SC. When indirect SC was applied in a study arm, 57% (n = 12) used video, 14% (n = 3) used radio, 10% (each n = 2) used reading comic books or reading a story, and 5% (each n = 1) used participatory theatre or Photo Voice. Details about each SC intervention can be found in **Table 1**.

**Stigmas targeted.**    The stigma categories targeted concerned mental health, physical health and not health-related stigmas were targeted (see **Fig 2**). Mental health was considered most frequently (n = 19, 43%), including general/multiple mental health conditions (n = 13, 68%), schizophrenia (n = 3, 16%), depression, autism and obsessive-compulsive disorder (each n = 5, 2%). Physical health stigmas (n = 13, 30%) concerned HIV/AIDS (n = 8, 62%), albinism (n = 2, 15%), general disabilities, leprosy, and diabetes mellitus type 1 (each n = 1,82%). Not health-related stigmas targeted (n = 11, 23%) included Lesbian, Gay, Bisexual, Transgender, Queer, Intersex, Asexual, + persons (LGBTQIA+) (n = 5, 46%), age (n = 2, 18%), refugee

**Table 1. Main table with study characteristics.**

| Reference | Country | Study design (study arms)[1] *Follow-up if conducted* | Target stigma | Target population (Social Contact setting) | Mean age (SD), age range | SC type and description *(Duration SC; Duration intervention if different)* *+ other intervention components (if applicable)* | Cultural adaptation[3] | Effectiveness stigma-related outcomes[4]: Significance; effect size if provided | Overall quality appraisal[5] |
|---|---|---|---|---|---|---|---|---|---|
| | | | | | **Indirect Contact–Main intervention** | | | | |
| | | | | | *Mental health* | | | | |
| Arthur et al., 2020a (Arthur et al., 2020b) [51,52] | Ghana | RCT (2) *3 months* | Depression, schizophrenia | Community leaders (community spots) | 40–49 (modal age), NR | Video of real-life experiences of a person with depression (NR;3hrs) + *Education* | Yes (2,4) | Personal stigma**S (T)**; NS (G); NS (I); 0.16*d* (I) Perceived stigmaNS (T), **S (G)**, **S (I)**; 0.17*d* (I) Social distance**S (T)**; NS (G); NS (I); 0.04d (I) Benevolence **S (T)**; NS (G); **S (I)**; 0.26d (I) Social restrictiveness**S (T)**; NS (G); NS (I); 0.46*d* (I) Community Mental Health Ideology**S (T)**; NS (G); **S (I)**; 0.60d (I) | Moderate (moderate) |
| Finkelstein et al., 2008 [53] | Russia | RCT (3) *6 months* | Schizophrenia | Students (higher education) | 18.9 (1.4) to 19.3 (1.6), NR | Story of a person with schizophrenia (NR;NR) + *Education* | NR | Social distance **S (T)** Community attitudes toward Mental Illness **S (T)** | Moderate |
| Gürbüz et al., 2020 [54] | Turkey | RCT (2) *6 months* | Obsessive-compulsive disorder | Community members (community spots) | 37.1 (13.3), 18–67 | Video in which PWLE and PWLE-family-member talk about life experiences (NR;NR) + *Education* (Control: similar but less stigmatised condition (MS)) | Yes (1) | Social distance **S (T, G, I)** 0.33η2 (T), 0.04η2 (G), 0.24η2 (I) Beliefs towards Mental Illness **S (T, G, I)** 0.34η2 (T), 0.09η2 (G), 0.29η2 (I) | Moderate |
| Maulik et al., 2019 (Maulik et al., 2017) [55,56] | India | NC (1) *22 months* | General | Community members (community spots) | 42 (15.7), 18–90 | Video of experiences of PWLE (NR;3months) + *Education* | Yes (1,4) | Barriers to Access to Care-TS **S (T)** Mental Health— KAP **S (T)** *Also qualitatively measured* | High (moderate) |
| Ng et al., 2017 [57] | Malaysia | NC (1) | General | Healthcare workers (health) | NR, 22–59 | Video with several elements, including filmed testimonies of PWLE and people close to PWLE, an interview with a successful person in recovery (5min) + *Education* | Yes (5) | Attitude **S (T)** Help-seeking behaviour **S (T)** Social distance **S (T)** | Moderate |

*(Continued)*

**Table 1.** (Continued)

| Reference | Country | Study design (study arms) [1] *Follow-up if conducted* | Target stigma | Target population (Social Contact setting) | Mean age (SD), age range | SC type and description *(Duration SC; Duration intervention if different)* + *other intervention components (if applicable)* | Cultural adaptation [3] | Effectiveness stigma-related outcomes [4]: Significance; effect size if provided | Overall quality appraisal [5] |
|---|---|---|---|---|---|---|---|---|---|
| Nistor et al., 2021 [58] | Romania | QE (2) | Autism | Students, teachers as supervisors (secondary school) | 17.02 (4.6), 14–18 | Contact-based Education; Guide, Testimonials (10min;3days) + *Education* (Control: Guide only) | NR | *Reported in %* | Low |
| Rong et al., 2011 [59] | China | QE (2) *2 weeks, 1;6 months* | Depression | Students (higher education) | 20.2 (0.7), NR | Video of a student with depression talking on related experiences (direct contact was encouraged, not monitored) (18min;20hrs in 10days) + *Education* | NR | Social distance S (I); 0.42*d* (I) | Moderate |
| Tergesen et al., 2021 [60] | Nepal | RCT (3) | Depression (study 1 and 2), Psychosis (study 2)) | Students (higher education) | Study 1: 21.0 (1.1), NR; Study 2: 19.6 (1.0), NR | Video with personal testimonies of service users (8min) | Yes (1) | Social distance *S*\* (G) Implicit Association Test *NS*\* (G) \**Also qualitatively measured* | Moderate—moderate* |
| | | | | | | *Physical health* | | | |
| Creel et al., 2011 (Rimal et al., 2018) [32,61] | Malawi | RCT (3) | HIV | Community members (community spots) | 34.9 (13.2), NR | Listening to radio diaries from PWLE (arm 1,2) (20min) Group discussion with radio diarist (direct; arm 2) (30min) | Yes (1,4) | Fear of contact S (G) (arm 1) Shame S (G) (arm 1) Blame & judgement S (G) (arm 2) Willingness to disclose NS (G) | Moderate (moderate) |
| De Groot et al., 2021a [62] | Tanzania | QE (2) | Albinism | Students (secondary school) | 16.35 (NR), 13–26 | Video of 5 persons with albinism talking about their lives (9-11min) | Yes (1) | Social distance *NS*\* (T), NS (I) \**Also qualitatively measured* | Moderate |
| De Groot et al., 2021b [63] | Tanzania | QE (2) *2/3 weeks* | Albinism | Community members (community spots) | 41,2 (NR) to 41.8 (NR), 18–94 | Radio drama (arm 1) (9-10min) Radio interview (arm 2) (9-10min) | Yes (1,4) | Community stigma (EMIC) *S*\* (T) Social distance *S*\* (T) | Moderate |
| | | | | | | *Not health-related* | | | |

(Continued)

**Table 1.** (*Continued*)

| Reference | Country | Study design (study arms) [1] *Follow-up if conducted* | Target stigma | Target population (Social Contact setting) | Mean age (SD), age range | SC type and description (*Duration SC; Duration intervention if different*) + *other intervention components (if applicable)* | Cultural adaptation [3] | Effectiveness stigma-related outcomes [4]: Significance; effect size if provided | Overall quality appraisal [5] |
|---|---|---|---|---|---|---|---|---|---|
| Banerjee et al., 2015 [64] | India | RCT (2) *3 months* | Social castes | Students (higher education) | 23,8 (1.7), NR | Video with a part where people from the low castes narrated their experiences of being Dalit or low caste (30min) + Education | Yes (1,4) | Implicit Association Test **S** (T) | Moderate |
| Logie et al., 2021 (Logie et al., 2022) [38,65] | Uganda | NC (1) *8 weeks* | Sexual violence | Healthcare workers; Refugee youth (refugee settlement) | NR, 16–24 | Reading comic books (4 hours[2];1day) + Education | Yes (1) | Sexual violence stigma **S** (T) *\*Also qualitatively measured* | Moderate (moderate) |
| Pufahl et al., 2021 [66] | India | NC (1) | LGTBQI+ | Community members; PWLE (theatre) | NR (18+) | Face-to-face dialogue among students and elderly (90min) | Yes (1) | Attitudes **S (T)** | Moderate |
| | | | | **Direct Contact–Main intervention** | | | | | |
| | | | | *Mental health* | | | | | |
| Ahuja et al., 2017 [67] | India | NC (1) *1 week* | General | Students (higher education) | NR, 18–21 | Face-to-face interaction between panellist and group of students (NR;2hrs) + Education | Yes (3,4) | Community Attitudes toward Mental Illness **S** (T); 0.22–0.28 η2 (T) | Moderate |
| Fernandez et al., 2016 [68] | Malaysia | RCT (2) *1 month* | Mental health condition | Students (higher education) | 21.0 to 21.1, 20–23 | Personal testimony of person in recovery of mental health condition for a group of students (arm 1) (45min-135min) Watching a video of person living with mental health condition (indirect: arm 2) (40min;130min) + Education | NR | Attitude, Disclosure & Help-seeking, Social Distance **S** (T); 0.49η2 (T) NS (G, I); 0.04η2 (G), 0.05η2 (I) | Moderate |
| Hofmann-Broussard et al., 2017 [69] | India | QE (2) | Psychosis, depression | Healthcare workers (health) | NR, NR | Face-to-face interaction with a community member who had recovered from mental health condition (NR;4days) + Education | Yes (2) | Stigma S* (T) | Moderate |

(*Continued*)

**Table 1.** (Continued)

| Reference | Country | Study design (study arms) [1] *Follow-up if conducted* | Target stigma | Target population (Social Contact setting) | Mean age (SD), age range | SC type and description *(Duration SC; Duration intervention if different)* + *other intervention components (if applicable)* | Cultural adaptation[3] | Effectiveness stigma-related outcomes[4]: Significance; effect size if provided | Overall quality appraisal[5] |
|---|---|---|---|---|---|---|---|---|---|
| Ran et al., 2022 [70] | China | RCT (3) *3;9 months* | Schizophrenia | Family caregivers (community spots) | 59-8-60.8 (12.9–13.6), 18–75 | 12-session peer group including psychoeducation (4); enhanced contact single family (4) and enhanced contact peer families (4). (6hr;18hr) + education | NR | ASSS: Affiliate self-stigma scale (KAP) S* (T) | High |
| Shah et al., 2015 [71] | India | NC (1) | General | Healthcare workers (health) | 37, 20–57 | One-to-one interaction with service users (NR; 1wk) + Education | NR | Attitudes S (T) | Moderate |
| Vaghee et al., 2018 [72] | Iran | RCT (3) *1 month* | Schizophrenia, bipolar 1 disorder, acute depression | Students (higher education) | 22.1 (1.6), NR | Contact-based education: Personal testimonies of PWLE (3hrs in 3 days) (Arm 2: apprenticeship with interviews PWLE) | NR | Empathy S* (T) | Moderate |
| *Physical health* | | | | | | | | | |
| Shah et al., 2014 [73] | India | QE (2) | HIV | Students (higher education) | 19 (median), 18–29 | Personal testimony with Q&A (2hrs) + Education | Yes (2a) | Endorsement of coercive measures S* (T) Blame S* (T) Intention to discriminate S* (T) | Moderate |
| Wu et al., 2008 [74] | China | QE (2) *3;6 months* | HIV | Healthcare workers (health) | 35.4 (8.0), NR | Testimony by 2 HIV advocates (NR;5hrs) + Education | Yes (1,2) | Attitude/behaviour S (T) | Moderate |
| *Not health-related* | | | | | | | | | |
| Ahuja et al., 2019 [75] | India | RCT (2) | LGTBQIA+ | Students (higher education) | NR, 18–21 | Face-to-face interaction between person recovered from mental health condition and group of students (50min;2hrs[2]) + Education (Control: contact only) | Yes (1,3,4) | Attitude S* (T); 0.49d (T) Empathy S* (T); 0.42d (T) | Moderate |

*(Continued)*

**Table 1.** (Continued)

| Reference | Country | Study design (study arms) [1] *Follow-up if conducted* | Target stigma | Target population (Social Contact setting) | Mean age (SD), age range | SC type and description (Duration SC; Duration intervention if different) + *other intervention components (if applicable)* | Cultural adaptation [3] | Effectiveness stigma-related outcomes [4]: Significance; effect size if provided | Overall quality appraisal [5] |
|---|---|---|---|---|---|---|---|---|---|
| Ozaydin et al., 2021 [76] | Turkey | RCT (2) | Refugee | Nursing students (higher education; refugee centres) | 21.71 (0.72), NR | Training migration issues + practice on refugee health (9*16hrs; 12wks total) +*Education* | Yes (1) | Xenophobia S (T, G, I); 0.33η2 (T), 0.215 (G); 0,404 (I) Attitude towards Refugee Scale S (T, G, I); 0.044–0.328η2 (T), 0.056–0.232η2 (G, 0.078–0.271η2 (I) | Moderate |
| Pekçetin et al., 2021 [77] | Turkey | RCT (2) | Age | Students; Community-dwelling elderly (higher education; elderly home) | 20.11 (1.25) to 20.48 (1.15), 18–23; 74.66 (8.02), NR | Face-to-face dialogue among students and elderly (2hrs/wk,*8; same +45min) + *Education* | NR | Ageist attitudes *S** (T), NS (G, I) Helping attitudes *NS** (T), NS (G, I); 0.014*d* (G) | Moderate |
| Sakalli et al., 2003 [78] | Turkey | QE (2) | LGTBQIA+ | Students (higher education) | NR, 19–26 | Personal testimony and Q&A (45min) | NR | Attitudes S (T, I) | Moderate |
| Schloegel et al., 2016 [37] (only study 1) | China | QE (2) | Age | Employees (work site) | 33, 23–57 | Presentations by older employees to younger employees on software development (6hrs) + *Education* | Yes (1) | Bias in developer performance expectations S (I); 0.05η2 (I) Bias in general performance expectations S (I); 0.05η2 (I) Bias in general performance expectations NS (I); 0.02η2 (I) | Moderate |
| | | | | | | *Various* | | | |
| Bagci et al., 2020 [79] | Turkey | QE (3) | Various | Students (higher education) | 21.07 (1.50) to 21.63 (1.30), NR | Direct: Human Library event with interactions between a "book" (PWLE) and a "reader" (participant) (20-40min) | NR | Affective outgroup NR (T, I); 0.06–0.26η2 (T, I) Behavioural intention S (T, I); 0.00–0.11η2 (T, I) | Moderate |
| | | | | | | **Collaborative Contact–Main intervention** | | | |
| | | | | | | *Mental health* | | | |
| Kohrt et al., 2020 (Rai et al., 2018) [80,81] | Nepal | NC (1) *4;16 months* | Mental health condition | Healthcare workers; PWLE; People close to; Researchers (health) | NR, 20–50 + | Recovery stories including Q&A, collaborative activities (65hrs;10days) + *Education; Popular Opinion Leaders* | Yes (1,3) | Social distance S (T) Attitudes S (T) *Also qualitatively measured* | Moderate (moderate) |
| | | | | | | *Physical health* | | | |

(*Continued*)

**Table 1.** (*Continued*)

| Reference | Country | Study design (study arms) [1] *Follow-up if conducted* | Target stigma | Target population (Social Contact setting) | Mean age (SD), age range | SC type and description (*Duration SC; Duration intervention if different*) + *other intervention components (if applicable)* | Cultural adaptation [3] | Effectiveness stigma-related outcomes [4]: Significance; effect size if provided | Overall quality appraisal [5] |
|---|---|---|---|---|---|---|---|---|---|
| Apinundecha et al., 2007 [82] | Thailand | QE (2) | HIV | PWLE; People close to; Community members (community spots) | 40.8 to 44.7, NR | Community members and PWLE together develop a stigma reduction intervention (NR;8months) + *Education; Empowerment* | Yes (1,3,4) | HIV/AIDS stigma **S** (T) | Moderate |
| Chidrawi et al., 2016 (Chidrawi et al., 2014; French et al., 2014; French et al., 2015) [83–86] | South Africa | NC (1) *3;6 months* | HIV | PWLE; People close to, Community members, Spiritual leader (community spots) | 37, 27–52 | Community members and PWLE together develop a stigma reduction intervention (NR;1.5months) + *Education* | Yes (1, 2a) | Stigma experiences of PLWH NS (T); 2.51*d* Stigma for community **S** (T); 0.11–0.22*d* *Also qualitatively measured* | Moderate (moderate/ moderate/ moderate) |
| Doostri-Irani et al., 2017 [87] | Iran | NC (1) | Diabetes Mellitus type 1 | PWLE; People close to; Community members; Healthcare workers (community spots; health) | NR, 18–94 | Community members and PWLE and healthcare workers together develop a stigma reduction intervention (NR;3yrs) + *Education; Empowerment; Advocacy; Protest* | Yes (1) | *Only qualitatively measured* | Low |
| Jain et al., 2013 [88] | Thailand | NC (1) | HIV | Community members (community spots) | 43.0, 15+ | Buddy pair (PWLE and non-PWLE) who do all kinds of activities together (NR;1yr) + *Education* | Yes (1) | Fear of HIV **S** (T) Social judgement **S** (T) | Moderate |
| Prinsloo et al., 2016 [89] | South Africa | NC (1) | HIV | Community members; PWLE (community spots) | NR, NR | Community-based stigma reduction activities and develop together a stigma reduction intervention (NR;5months) + *Education; Empowerment* | Yes (1) | *Only qualitatively measured* | Moderate |
| Uys et al., 2009 [90] | Lesotho, Malawi, South Africa, Swaziland, Tanzania | NC (1) | HIV | Healthcare workers; PWLE (health) | 37.9 (8.8), NR | PWLE and healthcare workers together develop a stigma reduction intervention (NR;1month) + *Education; Empowerment; Popular Opinion Leaders* | Yes (1) | HASI-PLWHA NS (T) HASI-Nurses **S** (T) *Also qualitatively measured* | Moderate-low* |

(*Continued*)

**Table 1.** (Continued)

| Reference | Country | Study design (study arms) [1] *Follow-up if conducted* | Target stigma | Target population (Social Contact setting) | Mean age (SD), age range | SC type and description *(Duration SC; Duration intervention if different)* + *other intervention components (if applicable)* | Cultural adaptation [3] | Effectiveness stigma-related outcomes [4]: Significance; effect size if provided | Overall quality appraisal [5] |
|---|---|---|---|---|---|---|---|---|---|
| colspan over | | | **Imagined Contact–Main intervention** | | | | | | |
| | | | *Physical health* | | | | | | |
| Carvalho-Freitas et al., 2017 [91] | Brazil | RCT (2) | Disabilities | Study 1: students (higher education); Study 2: Employees (work site) | Study 1: 21.38 (2.82), 17–36; Study 2: 32.09 (9.19), 19–67 | Imagining contact of yourself working with a person with a disability (3min) | NR | <u>Belief performance</u> Study 1: **S** (G); (0.02η2) (G) Study 2: **S** (G); (0.03η2) (G) <u>Expected work</u> Study 1: **S** (G); (0.02η2) (G) <u>Support rights</u> Study 2: **S** (G); (0.04η2) (G) | Low |
| | | | *Not health-related* | | | | | | |
| West et al., 2015 [92] *(only study 2)* | Jamaica | QE (3) | LGTBQIA+ | Students (higher education) | 21.4 (5.2), NR | Imagining contact of meeting a gay male stranger for the first time (arm 1) (5min) Imagining contact of meeting a gay male man with priming conditions (arm 2) (5min) | NR | <u>Attitude</u> **S**\* (G) <u>Social acceptance</u> *NS*\* (G) | Moderate |
| | | | **Vicarious Contact–Main intervention** | | | | | | |
| | | | *Not health-related* | | | | | | |
| Tercan et al., 2021 [93] | Turkey | QE (2) | Nationality (Syrian) | Children (primary school) | NR, 8–9 | Reading stories about interaction between a Syrian child and Turkish child (4hrs in 6 weeks) | NR | <u>Helping intentions</u> *NS*\* (G); 0.01η2 \**Also qualitatively measured* | Moderate-moderate\* |
| | | | **Main interventions with combinations of social contact types** | | | | | | |
| | | | *Mental health* | | | | | | |
| Altindag et al., 2006 [94] | Turkey | QE (2) *1 month* | Schizophrenia | Students (higher education) | 19.5 (1.0) to 19.7 (1.0), 18–23 | Direct: Face-to-face interaction between young person with schizophrenia and group of students Indirect: Autobiographical movie of a person with schizophrenia (6 hrs[2];1day) + *Education* | NR | <u>Social distance</u> *S*\* (T) <u>Attitude</u> *S*\* (T) | Moderate |

*(Continued)*

**Table 1.** (Continued)

| Reference | Country | Study design (study arms)[1] *Follow-up if conducted* | Target stigma | Target population (Social Contact setting) | Mean age (SD), age range | SC type and description *(Duration SC; Duration intervention if different)* + *other intervention components (if applicable)* | Cultural adaptation[3] | Effectiveness stigma-related outcomes[4]: Significance; effect size if provided | Overall quality appraisal[5] |
|---|---|---|---|---|---|---|---|---|---|
| Duman et al., 2017 [95] | Turkey | QE (2) | Schizophrenia | Students (higher education) | 21.1 (1.0) to 21.9 (1.0), NR | Indirect: Video with a part where people from the low castes narrated their experiences of being Dalit or low caste person Direct: Clinical practice where there is contact with psychiatric patients (NR;15days) + *Education* | NR | Beliefs towards Mental Illness $S^*$ (T, G) | Moderate |
| Kohrt et al., 2021 (Kaiser et al., 2022) [96, 97] | Nepal | RCT (2) *4;16 months* | General | Healthcare workers (health) | 36.2 (8.8), 21–56 | Indirect: Photographic narratives called "Photo Voice"; Direct: Face-to-face interaction as PWLE co-facilitated mhGAP training (NR;3months of which 9days mhGAP) + *Education; Popular Opinion Leaders* | Yes (1,2,3) | Social distance $S^*$ (T) Attitudes $S^*$ (T) *Also qualitatively measured* | Moderate (high) |
| Zhang et al., 2022 [98] | China | RCT (2) *1;3 months* | Bipolar, schizophrenia | Healthcare workers (health) | 36.7–39.85 (7.61–7.71), NR | Direct: 2/3 PWLE share experiences of recovery. Dialogue encouraged. Indirect: the recovery stories (of the present PWLE) are played. (1hr; 3hrs) + *Education* | Yes (1) | MICA: Mental Illness-Clinician's Attitudes **S** (T, I) RIBS: Reported and Intended Behaviour Scale **S** (T, I), NS (G) | Moderate |
| *Physical health* | | | | | | | | | |
| Dadun et al., 2017 (Peters et al., 2015; Peters et al., 2016) [43,99,100] | Indonesia | RCT (4) | Leprosy | PWLE; community members (community spots) | 36.5 to 42.2, NR | Indirect: Reading comic books and watching participatory videos made by PWLE depicting their life experiences (arm 1, 2) Direct: Face-to-face interaction/dialogue between PWLE and community members (arm 1, 2) (NR;2yrs) + *Counselling (arm 1)* + Socio-economic *support (arm 2)* | Yes (1,4) | SARI Stigma Scale (PWLE) $S^*$ (T) Participation Scale (PWLE) $S^*$ (T) Community stigma (EMIC) $S^*$ (T) Social distance $S^*$ (T) *Also qualitatively measured* | Moderate (moderate/high) |

(*Continued*)

**Table 1.** (Continued)

| Reference | Country | Study design (study arms)[1] *Follow-up if conducted* | Target stigma | Target population (Social Contact setting) | Mean age (SD), age range | SC type and description (Duration SC; Duration intervention if different) + *other intervention components (if applicable)* | Cultural adaptation[3] | Effectiveness stigma-related outcomes[4]: Significance; effect size if provided | Overall quality appraisal[5] |
|---|---|---|---|---|---|---|---|---|---|
| | | | | *Not health-related* | | | | | |
| Logie et al., 2019 [101] | Lesotho, Swaziland | NC (1) | LGTBQI+ | Students; healthcare workers; community members; PWLE; educators; police; community leaders (theatre) | NR (18+) | Indirect: Participatory theatre (2hrs) Direct: Audience participation in skits | Yes (1) | *Only qualitatively measured* | Moderate |

[1]RCT = Randomised Controlled Trial; QE = quasi-experimental study; NC = non-comparison (single-arm) intervention study

[2]Approximate duration indication, estimated by researchers.

[3]Cultural adaptation: (1) the intervention was at least partially originated from the local context; (2) the intervention was pre-tested/piloted/field-tested, or (2a) the intervention was piloted but it was unclear how this was done; (3) local beliefs, perceptions, and/or myths were taken into account; (4) local customs, cultural norms, resources, and/or habits were used to embed the intervention; and (5) translation of the intervention only. Built on Clay et al., 2020.

[4]The effects were reported as follows: **S** = ~properly reported + significant; NS = ~properly reported + not significant; *S** = incompletely reported + significant; *NS** = incompletely reported + not significant. (T) = main effect of Time; (G) = main effect of Group; (I) = Interaction effect. If reported, effect sizes were given. *d* of 0.2 is considered as small, 0.5 as medium, and 0.8 as large (Cohen's d). partial $\eta^2$ of 0.01 is considered as small, 0.06 as medium, and 0.14 as large (Cohen's F). R2 of 0.02 is considered as small, 0.13 as medium, and 0.26 as large ($F^2$ rules of thumb).

[5] See **S3 Text** for more details on the quality appraisal. In case of other publications next to the main publications, quality is given in brackets. When a study used mixed methods, the quality of the quantitative as well as the qualitative part (marked with a *) is given.

status, (Syrian) nationality, social castes, and having experienced sexual violence (each n = 1, 9%). Finally, one study (2%) targeted various health- and not health-related stigmas.

**Settings and target populations.** Studies were mostly conducted in one setting, typically in higher education (n = 14, 32%), at community spots (n = 11, 25%), or health settings (n = 8, 18%). Four studies (9%) were conducted in more than one setting (see **Fig 3**). Young adults were the target population of one-third of the studies (n = 15, 34%), and young adults together with adults were targeted in one-fourth (n = 12, 27%). Adults alone were the target group in ten studies (23%). Children together with young adults, and children alone were targeted in two studies each (5%), while one study (2%) targeted a combination of children, young adults and adults. Two studies (5%) did not report on age (see **Fig 4**).

**Stigma measures and measurements.** Different stigma-related measures were used. Most studies (n = 41, 93%) measured stigma quantitatively using a range of stigma scales. Nine of these studies (22%) complemented quantitative with qualitative measures, while few studies (n = 3, 7%) applied qualitative methods only to assess stigma, using open-ended question-naires, individual interviews and/or focus group discussions, reporting on any changes experienced after the SC intervention. Of the 44 studies, the majority (n = 36, 82%) measured stigma-related outcomes before and after the intervention. Of the 44 studies, eighteen studies (41%) performed single (n = 11, 25%) or multiple (n = 7, 16%) follow-up measurements,

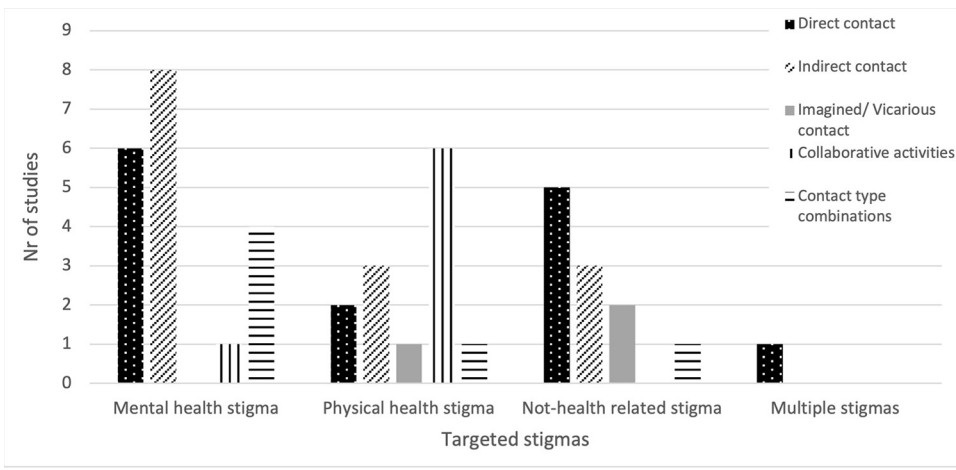

**Fig 2. Application of contact types per stigma category.**

which took place between 1 week to 22 months after the intervention. Of these, five (28%) concluded the last measurement within one month after study end and eleven (61%) after 6 months and beyond. Five studies (11%) measured changes–e.g. stigma or self-esteem–with PWLE.

**SC and other stigma reduction strategies.** The emphasis of SC in the intervention varied. In one-quarter of the studies (n = 12, 27%), SC was the main stigma reduction strategy. In the other studies (n = 32, 73%), SC was combined with at least one other strategy: education (n = 31, 97%), empowerment (n = 4, 13%), popular opinion leaders (n = 3, 9%), counselling, socio-economic support, advocacy, and protests (each n = 1, 1%).

**SC and intervention duration.** Overall intervention periods ranged from 3 minutes for an imagined contact intervention to 3 years for an intervention with collaborative activities.

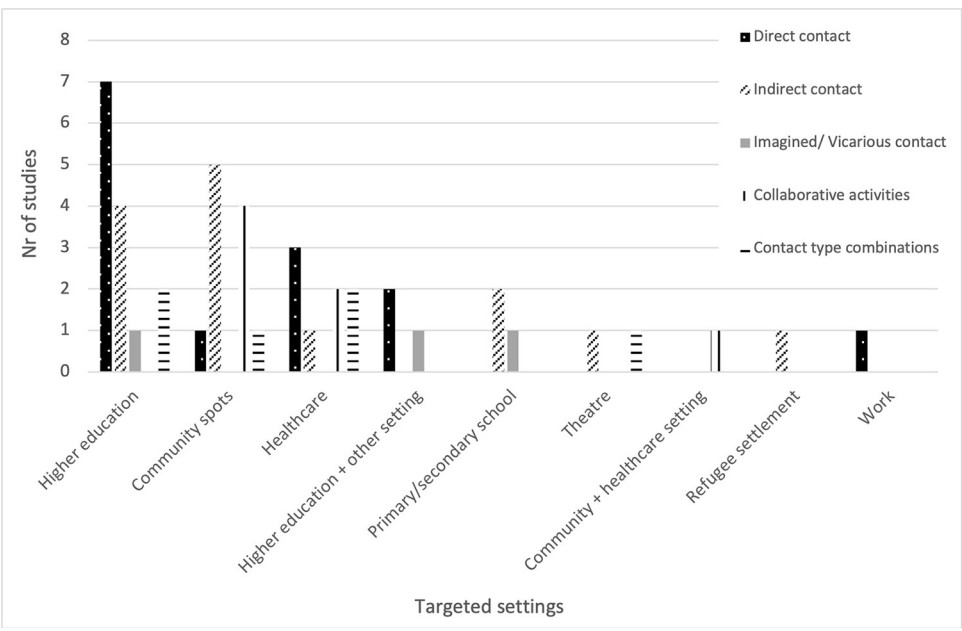

**Fig 3. Application of contact type per setting.**

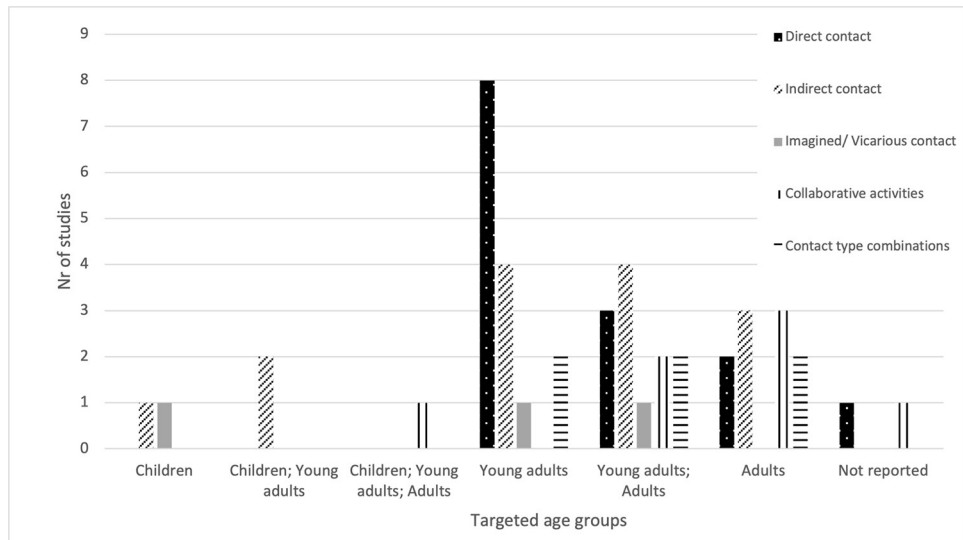

**Fig 4. Application of contact type per age group.**

Most interventions (n = 20, 45%) lasted less than one day, nine (20%) between one day and one month, nine (20%) between one month and one year, and three (7%) interventions took place for one year or more. Three studies did not report on intervention period. Direct, indirect, and imagined SC varied between 45 minutes to 65 hours, 5 minutes to 90 minutes, 3 minutes to 1 hour, respectively. One vicarious contact intervention had 6 sessions with 40 minutes of contact. Contact time within the studies employing collaborative activities (n = 7, 17%) was not computable.

**Choices for SC type.**   Almost half of the studies (n = 18, 41%) provided an explanation for their choice of SC type beyond the rationale for SC. About half of these studies (n = 8, 44%) employed indirect SC. We divided the choices into two categories: practical and contextual/cultural. As reasons of practicality, accessibility of the intervention or feasibility of use was mentioned most (n = 8, 44%), followed by financial resources (n = 7, 39%), potential for reach (n = 6, 33%) and the daily reality which may hamper contact in real life (n = 3, 17%). Time was mentioned once. Contextually, the cultural sensitivity around the stigma, such as its illegality, was considered most (n = 4, 22%), followed by the fit of the SC type with the population (n = 3, 17%).

**Intervention cultural adaptation.**   Two-third of the studies (n = 29, 66%) reported on (partial) cultural fit of the intervention. Of these studies, 86% (n = 25) mentioned the intervention at least *partially originated from the local setting*, nine (31%) referred to *using local customs* such as listening to the radio together in the comfort of someone's home, five (17%) indicated that the intervention was *pre-tested or piloted* or that *local beliefs were taken into account* such as considering context-related myths, and one (3%) indicated *adaptation consisted of translation*. In general, the studies reported minimally on the details of cultural adaptation. The type of adaptation per study can be found in **Table 1**.

**Positive factors for SC.**   One-fifth (n = 9, 20%) of the studies explicitly described the positive factors they considered when developing and/or implementing SC, and one-third (n = 16, 36%) did so implicitly. Overall, the studies concerning direct SC interventions or interventions combining two or more SC types applied, explicitly or implicitly, the most factors as, respectively, ten (71%) and three studies (60%) applied one or more factors. In imagined contact (n = 2) no positive factors were mentioned.

Of the studies integrating positive factors, the most employed factor was the creation of an interactive session (n = 9, 36%). This was followed by training of the resource person for the contact role, PWLE (moderately) disconfirming the stereotype and two of Allport's conditions, namely equal status and support by authorities (each n = 7, 28%). Five contact interventions indicated to embed the contact in the context, to include education and to create perspective and empathy before facilitating contact, the creation of a friendly environment, the focus on recovery (all n = 5, 20%) and ensuring that PWLE resource persons are similar to the audience (each n = 4, 18%).

**Lessons and recommendations to improve SC.**  About half of the studies (n = 24, 54%) shared learnings or recommendations regarding the application of SC. The positive factor mentioned most in these studies (n = 9, 38%) was to create multiple contact moments. Other highlighted recommendations were the training of resource persons (n = 6, 27%), support for behaviour change of the participants, focus on recovery, acknowledgement of potential risks for PWLE participation and the follow-up on expected moments and challenges during the contact process (each n = 4, 18%). Factors mentioned in three studies (14%) were ensuring high levels of intimacy, creating positive experiences, strengthening support from family and friends, and recognising the demand the contact role puts on the PWLE. One study mentioned that, despite encouragement, interaction between PWLE and the target group was limited.

In **Table 2**, we summarised the factors applied, and lessons learnt in order to strengthen SC, mentioned by the included studies.

**Effectiveness of SC interventions.**  Of all studies reporting on stigma reduction quantitatively (n = 41–excluding one study reporting in percentages only), almost all studies (n = 38, 95%) reported statistically significant (main) time, (main) group or interaction effects on at least one stigma measure. However, reporting was often incomplete and the performed statistical analysis was often inadequate (see explanation below), which means that no conclusive interpretations about effectiveness could be made. Effect sizes were reported in 14 studies (35%), indicating negligible to small effects across studies. **Table 1** includes details concerning reported outcomes per study.

In the measurement of *main effect of time*, nine studies (22%) did not report on this. Of the studies that did report on this (n = 32, 78%), two-thirds (n = 20, 63%) of the quantitative studies reported statistical significance; another eleven comparative studies (34%) reported invalid statistical significance as they did not compare the main intervention with the control arm(s). One study (9%) reported no statistically significant effect of time. The studies reporting a significant main effect of time included sixteen (84%) targeting mental health stigma, eight (73%) physical health stigma, seven (70%) not health-related stigmas and none (0%) multiple stigmas. While all five contact combination interventions and collaborative intervention studies reported a significant main effect of time, nine (64%) of the indirect contact interventions (n = 14) and none of the imagined or vicarious (n = 3) did. Of the comparative studies that could measure *main effect of group* (n = 31), more than half of the studies (n = 19, 61%) did not report on this. Of the studies that reported on *main effect of group* (n = 12, 39%), five (41%) reported statistically significant effects, three (25%) reported invalid statistically significant effects as they did not compare the main intervention with the control arm(s), and four (33%) reported no statistically significant effects. The studies reporting a significant main effect of group included four (29%) addressing mental health stigma, two (25%) physical health or not health-related stigmas and none (0%) multiple stigmas (n = 1). The interventions applying indirect contact (n = 10) reported, of all SC types, a significant effect of group (40%) most often. *Concerning interaction effect*, twenty (65%) of the comparative studies did not report on this. Of the eleven studies that did, eight (73%) showed statistically significant interaction effects hence stigma reduction, addressing mental health stigma (n = 4, 29% of all eligible

**Table 2. Positive factors applied and learned in included SC interventions (n = 44) as considerations.**

| Positive SC factors applied in included studies | Examples from interventions | Corresponding frameworks |
|---|---|---|
| **Contact Process** | | |
| PWLE and the target audience have equal status[7] | • Efforts to equalise relationships between HIV+ and HIV- facilitators [83]<br>• Role reversals to minimise power relations between doctor and patient [68] | [25] |
| Contact is supported by authorities or law[7] | • The intervention was conducted at the university, indicating that the institution was encouraging the event [79]<br>• Strongly acknowledged role of PWLE in the project by the organisation [89] | [25] |
| The different groups in contact share a common goal[4] | • The healthcare workers and the service users have a common goal, namely good services [80]<br>• The participants in this intervention are younger and older colleagues and have the same goals, namely creating business applications [37] | [25] |
| There is intergroup cooperation/no competition[4] | • Constructing a new context in which health service workers and service users plan (stigma reduction) activities together [90]<br>• Participants are encouraged to engage in respectful, positive intergroup contact [79] | [25] |
| The session is interactive/there is discussion[9] | • Using the principles of participation for collective learning [89]<br>• The promotion of group discussions [93] | [40] |
| Contact strategy uses 'pretend play' to make it less formal[1] | • This intervention turns the PWLE into 'books' and the target audience into 'readers', due to which both pretend to be something else [79] | |
| Frequent/multiple contact moments[3] | • This intervention uses multiple forms of social contact, namely testimonies and participatory videos [100]<br>• This intervention applies various forms of indirect contact, e.g. PWLE celebrities, a testimony of a person in recovery, and personal testimony of a colleague [57] | [102] |
| **Contact Atmosphere** | | |
| Contact is supported by high levels of intimacy[3] | • This intervention learned about the importance of meaningful contact between HIV+ and HIV-facilitators [89]<br>• This intervention ensured the groups were small for intimate, honest contact [86] | [103] |
| The contact takes place in a friendly/ informal setting[5] | • Where needed, this intervention was conducted in the home of the target audience, to make use of the comfort of the home [63]<br>• This intervention creates a story in which the participants are interacting positively and become friends [93] | |
| **Contact Content** | | |
| The contact is led and informed by the local context[5] | • This intervention has investigated 'what matters most' to the target audience (healthcare workers) and informed the strategy accordingly [80]<br>• This intervention has conducted an exploratory study to understand the context and make choices for strategies [100]<br>• This intervention integrated feedback on the quality of radio dialogues to modify the content [32] | |
| PWLE are presented as peers/humans instead of patients[2] | • This intervention elevated the visibility and status of service users, to be seen by healthcare workers as skilled members of society [80]<br>• This intervention emphasised the position of PWLE in their own right [67] | |
| The message concerns PWLE in recovery[5] | • This intervention included a community member who recovered from a mental health condition, and had vignettes describing similar themes of recovery [69]<br>• This intervention identified, through What Matters Most, that recovery is an important theme, which they included in their myth-busting [80] | [102,104] |
| **Perspectives: PWLE profile** | | |
| PWLE involved (only moderately) disconfirms the stereotype[7] | • This intervention included video clips of a PWLE who disconfirmed the stereotype of a person with albinism by having success [62]<br>• These interventions included realistic views of PWLE by including struggles [67,75] | [104,105] |
| PWLE are similar to the audience, e.g. age[5] | • This radio intervention connected men to a male PWLE and women to a female PWLE [32]<br>• These interventions ensured that the PWLE resource person were of the same age and socio-economic status as the target audience [67,75] | |
| **Perspectives: PWLE preparation** | | |

*(Continued)*

**Table 2.** (Continued)

| Positive SC factors applied in included studies | Examples from interventions | Corresponding frameworks |
|---|---|---|
| There is sufficient training for PWLE to take up the contact role[7] | • These interventions learned about the importance of training PWLE well, e.g. for handling projects [89] or for the resource role [68]<br>• These interventions prepared the PWLE to take up their role, e.g. through a Photovoice trajectory [80] or through a participatory video project [43] | [102] |
| PWLE are involved in designing, and contextualising the intervention[0] | • This intervention, the development of a comic book, was developed through a workshop with the designated youth [38]<br>• The stigma reduction activities in this intervention were co-designed and -implemented by PWLE [82] | |
| Support is ensured for PWLE to participate, e.g. family, friends[0] | • This intervention learned about the importance of involving e.g. family members to support the participation of PWLE [81] | |
| PWLE take up the contact role with other PWLE; peer groups[0] | • A research assistant to this intervention reflected that it is important that the participants (PWLE in this case) get to know each other [43]<br>• This study learned about the importance of meeting others (PWLE) in a similar situation [80] | |
| The contact method and other formats fit PWLE[0] | • This intervention reflected upon the importance that the physical conditions of the participants (PWLE) connect to the methodology chosen (video-making in this case) [43]<br>• This intervention ensured that the participants (PWLE; refugee youth) were inspired by the methodology chosen (comic books) [38] | |
| **Perspectives: PWLE monitoring/ evaluation** | | |
| Potential risks for PWLE are recognised and mitigated[0] | • This study reflected upon the risk that PWLE, by becoming 'books', risked being objectified [79]<br>• This intervention recognised the potential dangers that are connected to disclosure, and the videos were not broadcast in their own sub-districts [43] | |
| Experience of the contact role is monitored and evaluated[0] | • This study was evaluated with PWLE (but not about how support to them can be improved to take up that resource role) [79] | |
| There is proactive follow-up on challenges, unexpected moments[0] | • This study identified that there can always be unexpected moments, such as unexpected disclosures from the audience when other testimonies are shared [100]<br>• This study realised that the venue was a source of mistrust, due to which people close to PWLE were concerned and not always supportive of PWLE to participate [81] | |
| There is recognition for demands the contact role has on PWLE[0] | • This study recognised that the intensity of the intervention required much time and energy from PWLE, and that it can be demanding for them [83]<br>• This study learned that e.g. house chores got in the way of participation, and also led to drop-out [81] | |
| **Perspectives: preparation of the target audience** | | |
| There is perspective-taking/ educational material before contact[5] | • This intervention included an activity which was intended for the target audience to gain perspective on the constraints PWLE can face [75]<br>• This intervention included stories from aspirational figures (fellow healthcare workers) to strengthen contact [80] | [21,26] |
| There is motivation/reward to participate[2] | • This intervention included the intervention into daily activities that workers wanted to participate in [37]<br>This intervention learned that mandatory training can impact motivation. They learned about strengthening autonomy *to choose to stay* by including aspirational colleagues [80] | [104] |
| Participants are supported in their behaviour change[3] | • This intervention learned about the importance to teach skills to interact positively with PWLE [68]<br>• This intervention learned about the importance of adding techniques, e.g. community conversation, to improve contact [89] | [102] |
| Rules of engagement prior to participation[1] | • This intervention included a set of rules and regulations to be a 'reader' and engage with PWLE ('books') [79] | |
| **Perspectives: preparation of the implementer** | | |
| The implementer or facilitator models a person-first approach and gives a positive example[0] | • This intervention received the comment that 'the research assistant' (or facilitator) 'does not mind drinking from the same glass with me' [43] | [102] |

[#] the number of included studies (n = 44) that **applied** this specific factor in their intervention.

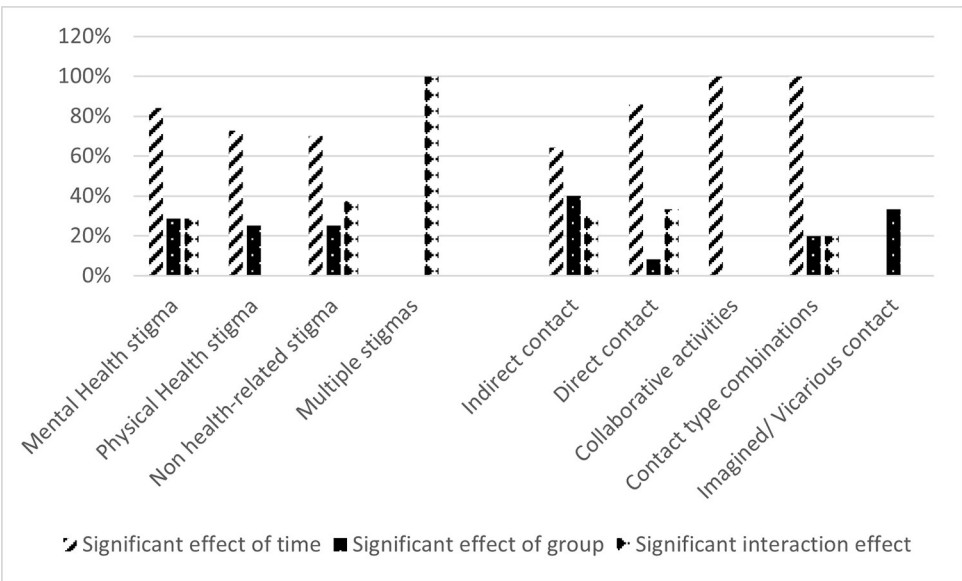

**Fig 5. Percentage of eligible included studies showing significance on time, group and interaction.**

mental health stigma studies), not health-related stigmas (n = 3, 38%) and multiple stigmas (n = 1, 100%). Three interventions (10%), of which two applied direct contact (one mental health and one non health related stigma) and one used indirect contact (physical health stigma), showed no statistically significant interaction effects. See **Fig 5** for an overview, per stigma category (left) and contact type (right), of the percentage of eligible studies reporting significance for a main effect of time and group, and interaction effects.

The eight interventions which showed a statistically significant interaction effect were conducted in Turkey (n = 4, 50%), China (n = 3, 38%) and Ghana (n = 1, 13%). Six (75%) combined SC with another stigma reduction strategy (education). Four of the interventions applied direct contact, demonstrating effectiveness in 25% of the interventions using direct contact (n = 12). Indirect contact was used in three interventions, showing effectiveness in 30% of the interventions applying indirect contact (n = 10). One of the five interventions combining SC types demonstrated effectiveness. See **Figs 6 and 7** for an overview of interaction effects per stigma category and social contact type, respectively. Two interventions (25%), one addressing multiple stigmas and the other a not health-related stigma, explicitly applied positive factors to improve SC, while the other interventions did so implicitly (n = 3, 38%) or did not mention it at all (n = 3, 38%). The SC component within the other interventions took between 18 minutes and six hours (n = 6, 75%), multiple days over a longer period (n = 1, 13%) or the duration was not reported (n = 1, 13%).

**Quality assessment according to JBI.** Of the 44 studies and their accompanying main publications, three (7%) was of low, thirty-nine (89%) of moderate and two (5%) of high quality (see **Table 1** and **S3 Text**). Although studies scored well on multiple aspects (see **S3 Text**), several points deserve additional attention. Within studies using a quasi-experimental design (n = 25, 57%), appropriate statistical analysis (n = 18, 72%) and completion of follow-up (n = 12, 48%) were limitedly reported. In studies applying a RCT design (n = 16, 36%), it was often unclear how different stages of blinding (n = 15, 94%), reliable outcome measurement (n = 15, 94%) and concealment of allocation (n = 8, 50%) were performed. None (0%) of the studies with qualitative methods (n = 6, 14%) reported on the position or influence of the

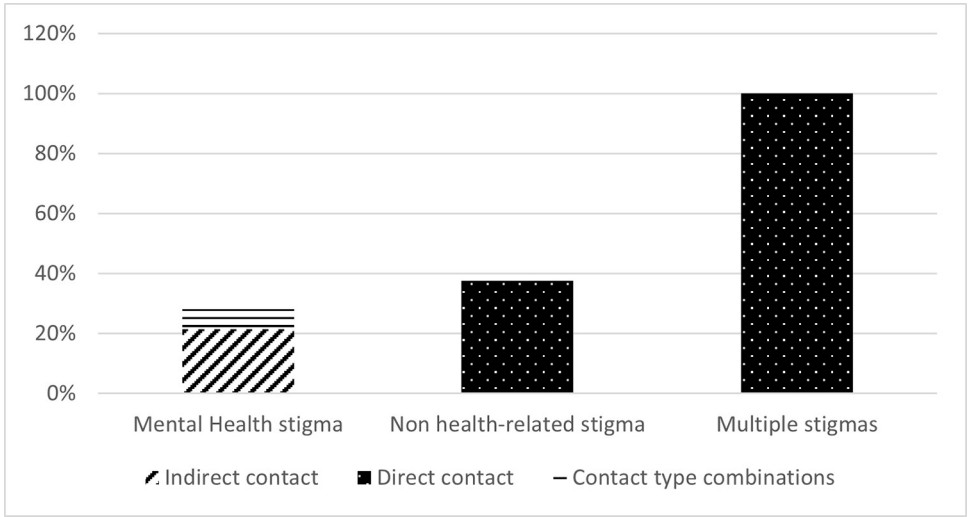

**Fig 6. Interaction effects per stigma category.**

researchers within the study. The eight statistically significant effective interventions were all of moderate quality.

## Explorative qualitative research

To explore expert perspectives and support and enrich the findings of the systematic review, we conducted six semi-structured individual interviews with stigma reduction researchers and/or practitioners with experience with SC strategies. Interviews lasted 42–54 minutes each. Of the respondents, four (67%) were female. Two respondents originally came from a LMIC. To avoid traceability and safeguard anonymity of respondents, no further demographic details are provided.

**Considerations for SC type.**   Some respondents mentioned that the context influenced the choice for a specific SC type to fit content and contact type with the target group. They

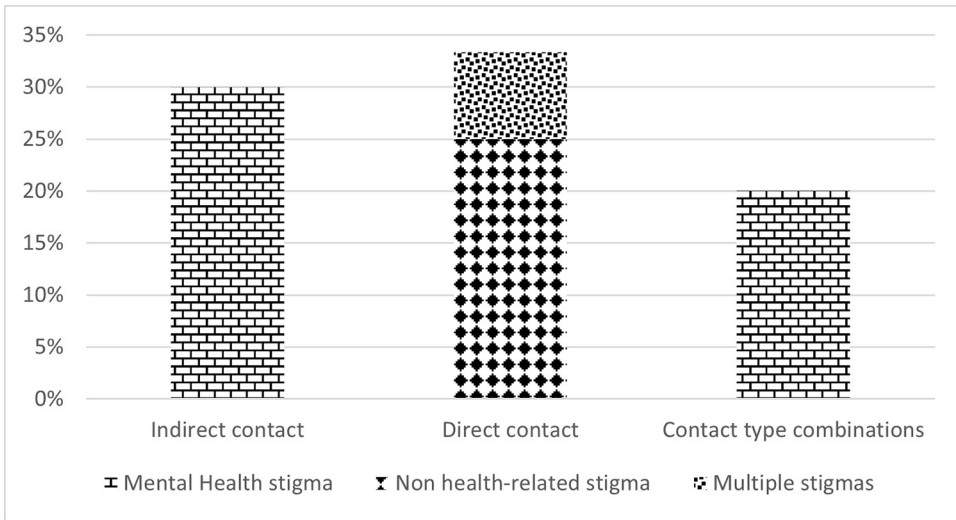

**Fig 7. Interaction effects per contact type.**

stated that they consciously chose to adapt the contact strategy and content into an engaging form for the target group. One interviewee argued that they consciously chose to apply imagined contact due to the conflictual nature of the setting. Others mentioned practical considerations such as lack of presence of PWLE and therefore the limited possibility to create direct contact or limited available resources like screens for showing video testimonials. Other context-related practicalities included costs, required permissions and available time of the implementing organisation.

**Contact in general: Content, process, atmosphere and sustainability.** All respondents emphasised the importance of carefully considering the context in which contact takes place, and the need to adapt SC to the context in content and process. They stressed that each situation and culture is different, with stigma experienced/expressed differently. The majority stated undertaking explorative studies to investigate the context to inform the development of the contact strategy was of major importance.

While a minority explicitly stated to have incorporated Allport's classic conditions for positive contact, almost all respondents referred to these factors to a certain extent. Institutional support and having a clear goal were predominantly mentioned. One interviewee indicated that they completely built their work on Allport's theory. Some respondents mentioned the importance of creating realistic contact scenarios and positive contact and to disconfirm stereotypes. Some respondents stated the importance of recognition with PWLE. One interviewee explicitly expressed the preference to create contact with peers instead of famous PWLE.

The majority argued that a good atmosphere contributes to the quality of contact. A few expressed the balance between informal (i.e. unstructured) and formal (i.e. structured) contact, and observed that informal contact moments such as having lunch (walks) and having fun together contributed to the quality of interaction. Another suggestion was to create small groups to ensure higher quality of contact.

When developing the contact strategy, the majority stressed to consider sustainability. Some noted that their studies focused predominantly on research, with sustainability not as a guiding issue. Nevertheless, one interviewee stressed that it would be useless to create an intervention which has no potential outside of research. The majority underlined to consider scaling the approach.

**Contact from the PWLE perspective: Preparation, participation, harm mitigation and monitoring.** All respondents emphasised the importance of considering the perspective of PWLE in the SC strategy. Some explicitly addressed these perspectives in their explorative studies to investigate their opinions, needs and views, while others had not incorporated PWLE perspectives but stated its importance.

There were some suggestions to empower PWLE before taking part in the contact strategy. Some respondents stressed that PWLE should feel comfortable to disclose and talk about the stigma. One interviewee argued they learned during intervention try-out that family involvement contributed to support of PWLE and advised to include this. A few explicitly mentioned to have trained PWLE beforehand, and stressed the importance:

*Yes, so a lot of preventive measures are there and then it goes on into an individual level, as well as to the family level as well. And then when I say individual level, that I mean, like during the training, the training sessions that I talked about, where they learn how to tell their stories, and how and all of those things, we also ended up, including, we also ended up including a lot of sessions on selfcare, you know, because when they're telling about their stories, most of them are telling stories that are traumatic to them.*

One respondent shared a dilemma to what extent PWLE may be instructed on what to say to the target audience to create the most impactful contact, as this instruction might limit PWLE in their freedom and autonomy to speak about their own experiences:

*. . . very often people with [condition] started talking about how difficult they were having it. And then I thought, no, that's, you know that, it's. . .. I'm very sorry, but, that's very bad and you should be able to share this, but if you want to change someone's view, and you are going to tell how bad everything is for you, then they don't think "oh, this is actually a human like you and me", actually that doesn't do that much.*

A frequently mentioned concern was to mitigate harm of PWLE, in line with their heightened vulnerability. One interviewee indicated that they encountered unexpected disclosure, and advised to be prepared for unexpected events when applying SC. The majority expressed a perceived risk of unintended consequences while employing SC, whereas only a minority indicated not having encountered or thought about such unintended consequences. Respondents were concerned and struggled with the notion that contact might increase stigma.

Some stated that it is imperative to evaluate the SC, whereas others did not reflect on this. Some respondents stressed the importance of after-care for PWLE or at least to check how PWLE have experienced the contact.

**Contact from the perspective of the target audience.** A few shared that it was also important to evaluate how target groups without the stigmatised characteristic had experienced the contact intervention and to check if they leave the intervention with the intended messages:

*It's just really important to evaluate as well, and to keep doing so. Because you just see a lot in contact interventions, with contact interventions there with people with [condition], that people with [disease] literally said: "no, but I can see everything just fine", and then people afterwards thought that that person would become blind. I don't know where they got that from, but that is very important to keep evaluating all the times.*

**Contact from the perspective of the implementing organization.** The challenge to motivate the implementing organisation to engage in contact was frequently mentioned. A common view amongst interviewees was that it was crucial to align with the wishes of the organisation, and to make sure that creating contact was something they sought for as well. In their experiences, this increased the motivation of other stakeholders to engage. Almost all respondents expressed the value of creating and cultivating good relationships with implementing organisations. Most suggested embedding the SC strategy within existing structures, such as existing classes at school, for greatest chance of success.

## Discussion

This paper provides an overview of SC stigma reduction interventions, employed across stigmas, populations and settings in LMICs, through a systematic review and expert perspectives. To the best of our knowledge, this is the first systematic review that summarises SC intervention research across stigmas and SC types.

This systematic review demonstrates that SC is a stigma reduction strategy applied across stigmas and settings, with almost half of the interventions addressing mental health stigma. The across-stigma application of SC supports recent calls to look beyond isolated stigmas in the development and implementation of stigma reduction strategies [12,39]. There is no

substantial discernible trend between stigma and SC type, although collaborative activities were foremost employed among physical health stigmas. Indirect and direct SC were mostly described in studies, while the more distant SC types, imagined or vicarious contact, were applied limitedly. Strikingly, none of the studies used online SC, also called E-contact, although interesting examples in HICs exist to bring people together online to reduce trans-gender stigma [106] and schizophrenia stigma [107]. Triggered by the worldwide Covid-19 pandemic, online SC could be an avenue to further explore, also in LMICs as internet accessi-bility is on the increase [108]. Children were underrepresented; echoing stigma reduction interventions in LMICs in general [15]. Another reason may be that a meta-analysis identified SC as more effective for adults than for children [109], which might have resulted in decisions not to apply SC among children. Another meta-analysis however concluded that imagined contact was more effective for children than adults and proposed imagined contact as a key component of child-focused education-based stigma reduction strategies [110]. Of the two studies in our review which targeted children specifically, Tercan et al. (2021) showed no sta-tistically significant stigma reduction [93] and Nistor et al. (2021) did not conduct statistical analysis [58]. We cannot draw conclusions on the effectiveness of the other studies targeting children next to (young) adults, as analysis was not age-stratified.

Most of the interventions were culturally adapted to a certain extent. This is key to ensure that interventions are relevant in the local context [16] and was identified as a core component for effective stigma reduction interventions [14]. While contrasting a recent scoping review in which only 20% of the included studies considered cultural values, meanings or practices [111], our finding confirms another recent review where half of the interventions were, to a certain extent, culturally adapted [14]. However, for most of the interventions, no or very few details were given how the intervention was made to align to the local context.

Although almost all SC interventions included in this review–across the various SC types and stigma categories–reported statistically significant stigma reduction and as such echo mul-tiple reviews highlighting SC as a promising and effective strategy [14,18,28]. This finding should be seen in the light that effectiveness was often reported inconsistently or incompletely. Additionally, while interventions using direct and indirect contact were most effective, this was only the case in about one-third of the interventions applying these SC types. Conclusions on effectiveness can only be drawn with a caveat. We found that the majority did not accu-rately report on time and/or group effects (i.e. time effects were not analysed irrespective of group and group effects were not analysed irrespective of time). Moreover, interaction effects were often not reported, although data to calculate these interaction effects were available. Additionally, we cannot rule out that studies that did not show positive results were all pub-lished [112]. The overall reported statistically significant stigma reduction might point to a risk of publication bias [14]. Altogether, this implies that the conclusions on effectiveness need to be viewed with caution, which is in line with a recent study that contested the evidence-base of mental health contact-based stigma reduction interventions [23]. The included studies which reported statistically significant interaction effects, consisting of interventions addressing men-tal health, not health-related and multiple stigmas and using direct, indirect and a combination of SC, were all of moderate quality, impeding the quality of evidence.

Our review has demonstrated that only few studies considered the perspective of PWLE in the SC intervention, and/or measured the effects of the SC interventions on PWLE, as explored recently [113]. This finding, confirming an earlier review on prejudice reduction [114], is striking as PWLE are key resource-persons in SC. It significantly contrasts with the idea of "nothing about us without us" [115]. In the recent Lancet Commission on ending stigma of mental health conditions, it is also emphasised that PWLE "need to be strongly supported to lead or co-lead interventions that use SC" [10]. Two important remarks can be made on the

base of our study. First, all experts within our qualitative research emphasised that PWLE should be meaningfully involved in developing and implementing the SC and stressed the importance of preparing, monitoring and evaluating SC with PWLE. Second, multiple of the included studies recommended how PWLE can be better prepared e.g. by involving family and friends, recognised the demand social contact can have on PWLE, or underlined the importance to monitor and evaluate with PWLE. This is supported by studies indicating that participating in SC can strengthen the social coping skills to deal with stigma, improve self-esteem and enhance personal empowerment of PWLE [36,113]; without losing sight of potential negative consequences [36].

We paid specific attention to the application of positive factors regarding applying SC. Strikingly, more than half of the included studies have not described if and how they embedded positive factors to improve their social contact intervention and the quality of the social contact, and of those that did, most did so implicitly. No conclusions on the impact of positive factors could be drawn. Several criticisms concerning positive factors have been mentioned, for example that, in the real world, ideal conditions for SC do not exist [116]. In our review, for example, some included studies argued that 'equal status' could not be created [79,80]. Importantly, it is the *quality* of contact that matters, rather than simply ticking the box of contact. One should be aware that facilitating SC does not necessarily result in positive interaction, and might even increase stigmatisation [24,36,42]. Participating in SC as PWLE might result in a more vulnerable position such as potential negative effects to self-disclosure [36]. This calls for careful reflection and development, together with PWLE, before bringing SC into practice.

Our study provides an overview of all factors applied and lessons learnt distilled from the included studies and interviews (see **Table 2**). These considerations are not exhaustive and directive for SC strategies, as contexts and realities differ. Rather, they should be seen as inspiration and guidance for critical reflection when considering SC.

This study includes the following strengths. First, it synthesises knowledge on SC used as stigma reduction strategy across stigmas, building on recommendations to identify cross-cutting features of stigma [12,39]. Second, extensive search methods were applied, contributing to thorough inclusion of literature. Third, we complemented the systematic review with expert perspectives. To our knowledge, this is a new methodological contribution within systematic reviews, and offers additional and in-depth insights. Lastly, during the data extraction and synthesis of the systematic review and the data analysis of the qualitative study, all data were analysed by two researchers to minimise subjectivity, which contributes to the reliability of this study.

Several limitations are recognised. First, we excluded studies targeting two-way prejudice, as it did not meet our stigma definition which is based in power. Second, we have only been able to interpret what has been reported, therefore we might have missed information when studies did apply positive factors but not reported upon it. The studies greatly varied in what they reported on SC strategy details. The explorative interviews mitigated this potential gap of knowledge. Moreover, we analysed the publication on what they explicitly, but also implicitly, reported on. Although these interpretations were checked by two researchers, it might be prone to interpretation errors. We therefore recommend future researchers to report more in detail about their intervention content and process. Third, we did not assess the validation process of the measures and did not explore the secondary benefits of effective stigma reduction thanks to SC, such as health impacts, as this was beyond the focus of this systematic review. Fourth, six first/corresponding authors were interviewed: only two came from a LMIC. Nonetheless, everyone worked from a specific LMIC context and worked in different types of SC across stigmas. As a final limitation, we did not interview PWLE.

## Conclusions

This study has provided an overview of SC stigma reduction interventions across stigmas, populations, and LMICs. Most of the interventions focused on mental or physical health-related stigmas and adult populations and applied foremost indirect and direct social contact. This review identified a challenge that effectiveness was often invalidly reported, overshadowing the conclusions that most interventions reported statistically significant stigma reduction. Therefore, while direct and indirect contact interventions showed the best results, no definitive statements can be made that any SC type is more effective than others. Similarly, no conclusions can be drawn that SC works better for a specific stigma category. To better understand the potential of SC as a stigma reduction strategy, we recommend 1) improving effectiveness reporting, including interaction effects and effect size and 2) including the under-reported effects of SC on PWLE. To understand the effects on children, we further recommend stratifying according to age. This review provides an overview of all included positive factors applied and lessons learnt to strengthen SC, which can be used as a set of considerations (adapted to each specific context) when developing and/or applying future SC to reduce stigma. We highly recommend future researchers to report in more detail on development, processes, content, positive factors and evaluation of SC strategies. Future SC research should pay attention to the controversies in the field. From an ethical perspective, participation of PWLE, as a key population in SC strategies, should be central to future research and SC strategies.

## Supporting information

**S1 Checklist. Inclusivity in global research.**
(DOCX)

**S2 Checklist. Supplementary material S1 Table: PRISMA checklist.**
(DOCX)

**S1 Text. Complete search strategy and inclusion/exclusion criteria.**
(DOCX)

**S2 Text. Topic guide.**
(DOCX)

**S3 Text. Details of quality appraisal according to JBI.**
(DOCX)

## Acknowledgments

We thank Dr Gabriela Koppenol-Gonzalez, Senior Researcher at War Child, for her support in assessing effectiveness.

## Author Contributions

**Conceptualization:** Carlijn Damsté, Kim Hartog.

**Data curation:** Carlijn Damsté, Kim Hartog.

**Formal analysis:** Carlijn Damsté, Kim Hartog.

**Methodology:** Carlijn Damsté, Kim Hartog.

**Project administration:** Carlijn Damsté, Kim Hartog.

**Supervision:** Kim Hartog.

**Validation:** Carlijn Damsté, Kim Hartog.

**Visualization:** Carlijn Damsté, Kim Hartog.

**Writing – original draft:** Carlijn Damsté, Kim Hartog.

**Writing – review & editing:** Carlijn Damsté, Petra C. Gronholm, Tjitske de Groot, Dristy Gurung, Akerke Makhmud, Ruth M. H. Peters, Kim Hartog.

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
