## [Decision Letter · Decision Letter 0]

5 Sep 2023

PGPH-D-23-00897

Social contact as a strategy to reduce stigma in low- and middle-income countries: a systematic review and explorative qualitative study

Dear Dr. Hartog,

Thank you for submitting your manuscript to PLOS Global Public Health. After careful consideration, we feel that it has merit but does not fully meet PLOS Global Public Health’s publication criteria as it currently stands. Therefore, we invite you to submit a revised version of the manuscript that addresses the points raised during the review process.

We look forward to receiving your revised manuscript.

Kind regards,

Anish Veshnal Cherian, Ph.D.

Academic Editor

Journal Requirements:

2. In the Funding Information you indicated that no funding was received. Please revise the Funding Information field to reflect funding received.

3. Please provide separate figure files in .tif or .eps format only and remove any figures embedded in your manuscript file. Please also ensure all files are under our size limit of 10MB.

Additional Editor Comments (if provided):

All the reviewer has recommended 5 to 7 comments hence requesting you to review and edit the manuscript accordingly.

Reviewers' comments:

Reviewer's Responses to Questions

**Comments to the Author**

1. Does this manuscript meet PLOS Global Public Health’s publication criteria? Is the manuscript technically sound, and do the data support the conclusions? The manuscript must describe methodologically and ethically rigorous research with conclusions that are appropriately drawn based on the data presented.

Reviewer #1: Partly

Reviewer #2: Yes

Reviewer #3: Yes

2. Has the statistical analysis been performed appropriately and rigorously?

Reviewer #1: N/A

Reviewer #2: Yes

Reviewer #3: I don't know

3. Have the authors made all data underlying the findings in their manuscript fully available (please refer to the Data Availability Statement at the start of the manuscript PDF file)?

Reviewer #1: Yes

Reviewer #2: Yes

Reviewer #3: Yes

4. Is the manuscript presented in an intelligible fashion and written in standard English?

Reviewer #1: Yes

Reviewer #2: Yes

Reviewer #3: Yes

5. Review Comments to the Author

Reviewer #1: This reviewer thought providing a high-level review focusing on broader conceptual and methodological issues might be helpful. I have stayed away from providing feedback section by section because it may be redundant if the major/broader issues in this manuscript are addressed.

Since stigma is linked to, among other factors, discrimination and social isolation of specific individuals, exploring the role of social contact in preventing or reducing the adverse effects of stigma on health outcomes is an essential contribution with significant theoretical, empirical, and practical implications. I commend the authors for engaging in such a significant and productive research study.

However, outlined below are some of the major conceptual and methodological issues that need to be addressed.

1. There is a need for clarity and perhaps consistency on the current study’s research question and broad objective/goal. In the abstract, the authors state, “Social contact (SC) has been identified as a promising strategy for stigma reduction.” The following study aim in line 32 of the abstract section says the current study is a systematic review that assesses the effectiveness of social contact (SC) as a stigma reduction strategy. However, the study goals in the introduction (see lines 111 to 116 of the introduction section) are inconsistent with the above-stated study aim. Instead, they read like objectives of a scoping review (I address this last aspect more in the next point.

2. Based on study objectives (see lines 111 to 116 of the introduction section) and the lack of specificity in outcomes of interests, this review should be a scoping review. For instance, no objective/question explicitly talks about the effectiveness of social contact in stigma reduction. As we say in research, the question drives the method. I also suggest that the authors review the research questions and methodologies used for different review types to make informed choices about whether to conduct a scoping review or revise the research questions to align with the research method—a systematic review.

3. Suppose the authors are interested in conducting a systematic review. In that case, they need to specify the aspect of stigma regarding the related health condition (i.e., mental health stigma, HIV/AIDS stigma, sexual violence). Choosing a specific type of stigma concerning a health condition allows the authors to make their research objectives specific enough to be measurable and testable. Although 41 studies found SC effective in reducing stigma, it is unclear how this effect is distributed across different health conditions (i.e., sexual violence, mental health, or HIV/AIDS).

4. Additionally, it enables the authors to discuss specific adverse impacts of stigma on a health condition and ways in which social contact (as opposed to, maybe, social isolation?) could mitigate those adverse effects or promote positive health outcomes in that particular condition. As of now, even the introduction section needs to be narrower.

5. In any review, it is important to discuss existing knowledge and gaps in existing reviews in the introduction to establish the relevance of the current study. I recommend that the authors of this systematic review consider adding that aspect to the introduction section in a paragraph that precedes the one on the study goal and research objectives.

6. Although this could be refreshingly innovative, it is still being determined why the authors used a mixed methods study that combines a systematic review with a qualitative inquiry. The rationale of this method needs to be explained in the introduction. Nor is it reflected in the research objectives in lines 111 to 116 of the introduction section. In other words, provide clarity on which research questions are addressed by the systematic review and which one will be qualitatively assessed. Additionally, when one uses a mixed methods study, it is essential to specify in the methods section which part of the mix is the major, which is the minor, and how they relate to each other.

7. Authors need an operational definition of social contact.

Reviewer #2: The study exhibited a well-conceived and meticulously executed research design. The team's endeavor to investigate the utilization of social contact strategies within low- and middle-income countries (LMICs) was both innovative and commendable, effectively addressing a critical knowledge gap for public health stakeholders.

The incorporation of a systematic review alongside interviews with the authors further bolstered the study's robustness and depth. This dual methodology approach contributed to the overall strength of the research.

Certainly, here are your suggestions rephrased in a professional manner for an academic review:

1. Regarding the Main Table: Consider reorganizing the study data in a chronological and, if feasible, regional order within the main table. This adjustment would facilitate a clearer understanding for readers of the geographical focus throughout the article.

2. Concerning the Results Section: Ensure consistency in the reporting of frequencies and percentages. Some instances display variations such as (n=0, 0%), 0(0%), or solely reporting 'n' or percentage values. Uniformity in this aspect would enhance the clarity of the findings.

3. Is there a specific reason why the authors did not include schizophrenia, depression, autism, and OCD within the category of mental health conditions in the study?

4. Regarding the Author's Inclusion of Literature in the Results Section: Clarify the rationale behind the inclusion of literature references within the results section (lines 349 to 355). This could be elaborated upon to justify its presence within this section.

5. Inconsistencies in Reference Style: Address inconsistencies in reference style observed in the manuscript (lines 349, 521, 526, 527, 532, 533, 535, 536, 550, and 590). A uniform reference style should be implemented throughout the document for consistency and adherence to academic conventions.

Reviewer #3: Great work by the authors team, writeup show academic fashion in research.

Some minor queries,

1. Multiple places its mentioned that 10% of the records were independently reviewed by both authors (Please crosscheck)

2.How did research team managed to review articles from 5 languages

3. This study can be splitted as two independent study, is there any specific reason to write such a huge paper.

4. In title (explorative qualitative study) is little confusing for the readers hence kindly think about it. (Social contact as a strategy to reduce stigma in low- and middle-income countries: a systematic review and exploration on author's perspective)

6. PLOS authors have the option to publish the peer review history of their article (what does this mean?). If published, this will include your full peer review and any attached files.

**Do you want your identity to be public for this peer review?** For information about this choice, including consent withdrawal, please see our Privacy Policy.

Reviewer #1: **Yes: **THABANI NYONI

Reviewer #2: No

Reviewer #3: No

---

## [Editor Report · Decision Letter 1]

9 Jan 2024

PGPH-D-23-00897R1

Social contact as a strategy to reduce stigma in low- and middle-income countries: a systematic review and expert perspectives

Dear Dr. Kim,

Thank you for submitting your manuscript to PLOS Global Public Health. After careful consideration, we feel that it has merit but does not fully meet PLOS Global Public Health’s publication criteria as it currently stands. Therefore, we invite you to submit a revised version of the manuscript that addresses the points raised during the review process.

We look forward to receiving your revised manuscript.

Reviewer -1 

This reviewer thought providing a high-level review focusing on broader conceptual and methodological issues might be helpful. I have stayed away from providing feedback section by section because it may be redundant if the major/broader issues in this manuscript are addressed. 

Since stigma is linked to, among other factors, discrimination and social isolation of specific individuals, exploring the role of social contact in preventing or reducing the adverse effects of stigma on health outcomes is an essential contribution with significant theoretical, empirical, and practical implications. I commend the authors for engaging in such a significant and productive research study.

However, outlined below are some of the major conceptual and methodological issues that need to be addressed. 

1. There is a need for clarity and perhaps consistency on the current study’s research question and broad objective/goal. In the abstract, the authors state, “Social contact (SC) has been identified as a promising strategy for stigma reduction.” The following study aim in line 32 of the abstract section says the current study is a systematic review that assesses the effectiveness of social contact (SC) as a stigma reduction strategy. However, the study goals in the introduction (see lines 111 to 116 of the introduction section) are inconsistent with the above-stated study aim. Instead, they read like objectives of a scoping review (I address this last aspect more in the next point.

2. Based on study objectives (see lines 111 to 116 of the introduction section) and the lack of specificity in outcomes of interests, this review should be a scoping review. For instance, no objective/question explicitly talks about the effectiveness of social contact in stigma reduction. As we say in research, the question drives the method. I also suggest that the authors review the research questions and methodologies used for different review types to make informed choices about whether to conduct a scoping review or revise the research questions to align with the research method—a systematic review. 

3. Suppose the authors are interested in conducting a systematic review. In that case, they need to specify the aspect of stigma regarding the related health condition (i.e., mental health stigma, HIV/AIDS stigma, sexual violence). Choosing a specific type of stigma concerning a health condition allows the authors to make their research objectives specific enough to be measurable and testable. Although 41 studies found SC effective in reducing stigma, it is unclear how this effect is distributed across different health conditions (i.e., sexual violence, mental health, or HIV/AIDS). 

4. Additionally, it enables the authors to discuss specific adverse impacts of stigma on a health condition and ways in which social contact (as opposed to, maybe, social isolation?) could mitigate those adverse effects or promote positive health outcomes in that particular condition. As of now, even the introduction section needs to be narrower.

5. In any review, it is important to discuss existing knowledge and gaps in existing reviews in the introduction to establish the relevance of the current study. I recommend that the authors of this systematic review consider adding that aspect to the introduction section in a paragraph that precedes the one on the study goal and research objectives. 

6. Although this could be refreshingly innovative, it is still being determined why the authors used a mixed methods study that combines a systematic review with a qualitative inquiry. The rationale of this method needs to be explained in the introduction. Nor is it reflected in the research objectives in lines 111 to 116 of the introduction section. In other words, provide clarity on which research questions are addressed by the systematic review and which one will be qualitatively assessed. Additionally, when one uses a mixed methods study, it is essential to specify in the methods section which part of the mix is the major, which is the minor, and how they relate to each other. 

7. Authors need an operational definition of social contact.

Decision – Major Revision 

Reviewer -2 

The study exhibited a well-conceived and meticulously executed research design. The team's endeavor to investigate the utilization of social contact strategies within low- and middle-income countries (LMICs) was both innovative and commendable, effectively addressing a critical knowledge gap for public health stakeholders.

The incorporation of a systematic review alongside interviews with the authors further bolstered the study's robustness and depth. This dual methodology approach contributed to the overall strength of the research.

Certainly, here are your suggestions rephrased in a professional manner for an academic review:

1. Regarding the Main Table: Consider reorganizing the study data in a chronological and, if feasible, regional order within the main table. This adjustment would facilitate a clearer understanding for readers of the geographical focus throughout the article.

2. Concerning the Results Section: Ensure consistency in the reporting of frequencies and percentages. Some instances display variations such as (n=0, 0%), 0(0%), or solely reporting 'n' or percentage values. Uniformity in this aspect would enhance the clarity of the findings.

3. Is there a specific reason why the authors did not include schizophrenia, depression, autism, and OCD within the category of mental health conditions in the study?

4. Regarding the Author's Inclusion of Literature in the Results Section: Clarify the rationale behind the inclusion of literature references within the results section (lines 349 to 355). This could be elaborated upon to justify its presence within this section.

5. Inconsistencies in Reference Style: Address inconsistencies in reference style observed in the manuscript (lines 349, 521, 526, 527, 532, 533, 535, 536, 550, and 590). A uniform reference style should be implemented throughout the document for consistency and adherence to academic conventions.

Decision – Minor Revision 

Reviewer – 3 

Great work by the authors team, writeup show academic inclination in research. 

Some minor queries,

1. Multiple places its mentioned that 10% of the records were independently reviewed by both authors (Please crosscheck)

2.How did research team managed to review articles from 5 languages

3. This study can be splitted as two independent study, is there any specific reason to write such a huge paper.

4. In title (explorative qualitative study) is little confusing for the readers hence kindly think about it. (Social contact as a strategy to reduce stigma in low- and middle-income countries: a systematic review and exploration on author's perspective)

Decision – Accept  

Kind regards,

Anish Veshnal Cherian, Ph.D.

Academic Editor

Journal Requirements:

Additional Editor Comments (if provided):

Dear Author,

Please address all the comments raised by the reviewers and submit the revised manuscript

Thank you
---

## [Editor Report · Decision Letter 2]

5 Mar 2024

Social contact as a strategy to reduce stigma in low- and middle-income countries: a systematic review and expert perspectives

PGPH-D-23-00897R2

Dear Kim,

We are pleased to inform you that your manuscript 'Social contact as a strategy to reduce stigma in low- and middle-income countries: a systematic review and expert perspectives' has been provisionally accepted for publication in PLOS Global Public Health.

Best regards,

Anish Veshnal Cherian, Ph.D.

Academic Editor

Dear Authors,

Thank you for the revised submission and we accept the revised version for publication.

Congratulations!!

best wishes